# Bilevel Distance Metric Learning for Robust Image Recognition

**Jie Xu**[1,2], **Lei Luo**[2], **Cheng Deng**[1,*], **Heng Huang**[2,3*]
[1] School of Electronic Engineering, Xidian University, Xi'an, Shaanxi, China
[2] Electrical and Computer Engineering, University of Pittsburgh, USA, [3] JDDGlobal.com
jie.xu@pitt.edu,leiluo2017@pitt.edu
chdeng.xd@gmail.com,heng.huang@pitt.edu

## Abstract

Metric learning, aiming to learn a discriminative Mahalanobis distance matrix $\mathbf{M}$ that can effectively reflect the similarity between data samples, has been widely studied in various image recognition problems. Most of the existing metric learning methods input the features extracted directly from the original data in the preprocess phase. What's worse, these features usually take no consideration of the local geometrical structure of the data and the noise that exists in the data, thus they may not be optimal for the subsequent metric learning task. In this paper, we integrate both feature extraction and metric learning into one joint optimization framework and propose a new bilevel distance metric learning model. Specifically, the lower level characterizes the intrinsic data structure using graph regularized sparse coefficients, while the upper level forces the data samples from the same class to be close to each other and pushes those from different classes far away. In addition, leveraging the KKT conditions and the alternating direction method (ADM), we derive an efficient algorithm to solve the proposed new model. Extensive experiments on various occluded datasets demonstrate the effectiveness and robustness of our method.

## 1 Introduction

Metric learning problem is concerned with learning an optimal distance matrix $\mathbf{M}$ that captures the important relationships among data for a given task, *i.e.*, assigning smaller distances between similar items and larger distances between dissimilar items. Generally, metric learning can be formulated as a minimal optimization about the objective function: $\mu\text{Reg}(\mathbf{M}) + \text{Loss}(\mathbf{M}, \mathcal{A})$, where $\text{Reg}(\mathbf{M})$ is a regularization term on the distance matrix $\mathbf{M}$ and $\text{Loss}(\mathbf{M}, \mathcal{A})$ is a loss function that penalizes constraints. Different choices of regularization terms and constraints result in various metric learning methods, *e.g.*, large-margin nearest neighbor (LMNN) [16], information-theoretic metric learning (ITML) [4], FANTOPE [7], CAP [5], *etc*. More recent works focus on using maximum correntropy criterion [18], smoothed wasserstein distance [19], matrix variate Gaussian mixture distribution [11] for metric learning formulations to improve the robustness. Although these methods achieve great success, they all mainly focus on improving the discriminability of the distance matrix $\mathbf{M}$ but ignore the discriminating power of input features. Especially, the descriptors of the sample pairs they address are usually extracted directly from the original data in the preprocess phase without considering the local geometrical structure of the data, thus such descriptors may not be optimal for the subsequent metric learning task.

Besides metric learning methods, many other machine learning tasks such as clustering and dictionary learning also suffer from the above limitation. To address this issue, the recently proposed solution

is to adopt the strategy of joint learning or bilevel model, and fortunately, great achievements have been made by many researchers. Wang *et al.* [15] propose a joint optimization framework in terms of both feature extraction and discriminative clustering. They utilize graph regularized sparse codes as the features, and formulate sparse coding as the constraint for clustering. Zhou *et al.* [23] present a novel bilevel model-based discriminative dictionary learning method for recognition tasks. The upper level directly minimizes the classification error, while the lower level uses the sparsity term and the Laplacian term to characterize the intrinsic data structure. Yang *et al.* [20] propose a bilevel sparse coding model for coupled feature spaces, where they aim to learn dictionaries for sparse modeling in both spaces while enforcing some desired relationships between the two signal spaces. All these models benefit from the joint learning strategy or the bilevel model and achieve an overall optimality to a great extent. Inspired by these works, we propose to extract features and learn the Mahalanobis distance matrix $\mathbf{M}$ through a unified joint optimization model.

How to choose the feature extraction model is also an important problem. Although metric learning task aims to learn a discriminative $\mathbf{M}$, it would be better if the input features are also of discriminating power. The common choice is principal component analysis (PCA) feature, which is able to reduce the data dimension and identify the most important features [1]. However, PCA feature is not necessarily discriminative and also may loss the useful information. More recently, sparse coefficients prove to be an effective feature which is not only robust to noise but also scalable to high dimensional data [17]. Furthermore, motivated by recent progress in manifold learning, Zheng *et al.* [22] incorporate the graph Laplacian into the sparse coding objective function as a regularizer, achieving more discriminating power compared with traditional sparse coding algorithms.

In this paper, we integrate the graph regularized sparse coding model into the distance metric learning framework and propose our new bilevel model. The lower level focus on detecting the underlying data structure, while the upper level directly forces the data samples from the same class to be close to each other and pushes those samples from different classes far away. Note that the input data samples of the upper level are represented by the sparse coefficients learnt from the lower level model. And benefiting from the feature extraction operation of the lower level model, the new features become more robust to noise with the sparsity norm and more discriminative with the Laplacian graph term. In addition, to solve our bilevel model, we transform the lower level problem of the proposed model into equality and inequality constraints and then apply ADM to solve it. Extensive experiments on various occluded datasets indicate that the proposed bilevel model can achieve more promising performance than other related methods.

**Notations**: Let $\mathbb{S}_+$ denotes the set of real-valued symmetric positive semi-definite (PSD) matrices. For matrices $\mathbf{A}$ and $\mathbf{B}$, denote the Frobenius inner product by $\langle \mathbf{A}, \mathbf{B} \rangle = Tr(\mathbf{A}^\top \mathbf{B})$, where '$Tr$' denotes the trace of a matrix. For a given vector $\mathbf{a} = (a_1, a_2, ..., a_d)^\top$, $diag(\mathbf{a}) = \mathbf{A}$ corresponds to a squared diagonal matrix such that $\forall i, A_{i,i} = a_i$. $\mathbf{e}_k \in \mathbb{R}^k$ represents a unit vector of length $k$, and $\mathbf{I}$ is a unit matrix. Finally, for $x \in \mathbb{R}$, let $[x]_+ = \max(0, x)$.

## 2 Bilevel Distance Metric Learning

### 2.1 Large Margin Nearest Neighbor

Let $\{(\mathbf{x}_1, y_1), ..., (\mathbf{x}_n, y_n)\} \in \mathbb{R}^d \times C$ be a set of labeled training data with discrete labels $C = \{1, ..., c\}$, where $n$ is the number of samples. Most of the metric learning methods aim to learn a metric, such as widely used Mahalanobis distance $d_{\mathbf{M}}(\mathbf{x}_i, \mathbf{x}_j) = \sqrt{(\mathbf{x}_i - \mathbf{x}_j)^\top \mathbf{M}(\mathbf{x}_i - \mathbf{x}_j)}$, to effectively reflect the similarity between data.

Large margin nearest neighbor (LMNN) [16], as one of the most widely used metric learning methods, requires the learned Mahalanobis distance to satisfy two objectives, *i.e.*, samples from the same class are forced to be close to each other and those from different classes are pushed far away. If we denote the similar pairs by $\mathcal{S}$ and triplet constraints by $\mathcal{T}$ as:

$$\begin{aligned} \mathcal{S} &= \{(\mathbf{x}_i, \mathbf{x}_j) : y_i = y_j \text{ and } \mathbf{x}_j \text{ belongs to the } k\text{-neighborhood of } \mathbf{x}_i\}, \\ \mathcal{T} &= \{(\mathbf{x}_i, \mathbf{x}_j, \mathbf{x}_k) : (\mathbf{x}_i, \mathbf{x}_j) \in \mathcal{S}, y_i \neq y_k\}, \end{aligned} \quad (1)$$

then LMNN model can be formulated as:

$$\min_{\mathbf{M} \in \mathbb{S}_+} (1 - \lambda) \sum_{(i,j) \in \mathcal{S}} d_{\mathbf{M}}^2(\mathbf{x}_i, \mathbf{x}_j) + \lambda \sum_{(i,j,k) \in \mathcal{T}} [1 + d_{\mathbf{M}}^2(\mathbf{x}_i, \mathbf{x}_j) - d_{\mathbf{M}}^2(\mathbf{x}_i, \mathbf{x}_k)]_+, \quad (2)$$

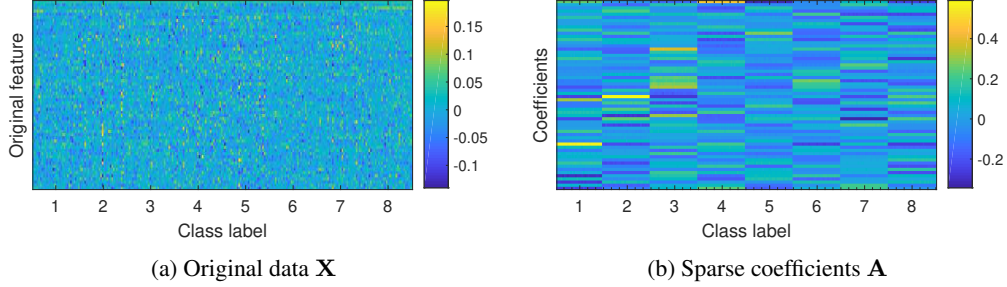

(a) Original data $\mathbf{X}$          (b) Sparse coefficients $\mathbf{A}$

Figure 1: Original data $\mathbf{X}$ and the corresponding sparse coefficients $\mathbf{A}$ learned from the proposed bilevel distance metric learning model. The x-axis represents different samples belonged to eight classes.

where $\lambda \in [0, 1]$ controls the relative weight between two terms. The objective function in Eq. (2) pulls the "target" neighbors whose labels are the same as $\mathbf{x}_i$'s toward $\mathbf{x}_i$ and pushes away the "impostor" neighbors whose labels are different from $\mathbf{x}_i$'s.

Although LMNN achieves good results, it learns the distance matrix to characterize the point-to-point distance which is sensitive to the noise. Furthermore, the descriptors of the sample pairs it addresses are usually extracted directly from the original data in the preprocess phase without considering the local geometrical structure of the data. Thus such features may not be optimal for the subsequent metric learning task. To address these problems, in this paper, we propose a bilevel model which jointly learns the distance matrix $\mathbf{M}$ and extracts features under a sparse representation-based framework.

## 2.2 Bilevel Distance Metric Learning

Sparse representations prove to be an effective feature for classification. Also, some researchers suggest that the contribution of one sample to the reconstruction of another sample is a good indicator of similarity between these two samples [3]. Thus the reconstruction coefficients can be used to constitute the similarity graph. Inspired by these findings, we integrate both sparse representation and graph regularization into a metric learning framework and propose our new bilevel distance metric learning model.

We assume all the data samples $\mathbf{X} = [\mathbf{x}_1, \mathbf{x}_2, ..., \mathbf{x}_n] \in \mathbb{R}^{d \times n}$ are represented by their corresponding sparse coefficients $\mathbf{A} = [\mathbf{a}_1, \mathbf{a}_2, ..., \mathbf{a}_n] \in \mathbb{R}^{k \times n}$ based on a learned dictionary $\mathbf{U} = [\mathbf{u}_1, \mathbf{u}_2, ..., \mathbf{u}_k] \in \mathbb{R}^{d \times k}$. Then the proposed bilevel distance metric learning model can be expressed as follows:

$$
\min_{\mathbf{M} \in \mathbb{S}_+^d, \mathbf{U}} \quad (1 - \lambda) \sum_{(i,j) \in \mathcal{S}} d_{\mathbf{M}}^2(\mathbf{a}_i, \mathbf{a}_j) + \lambda \sum_{(i,j,k) \in \mathcal{T}} [\xi + d_{\mathbf{M}}^2(\mathbf{a}_i, \mathbf{a}_j) - d_{\mathbf{M}}^2(\mathbf{a}_i, \mathbf{a}_k)]_+
$$

$$
s.t. \quad \mathbf{A} = \arg\min_{\mathbf{A}} \frac{1}{2}||\mathbf{X} - \mathbf{U}\mathbf{A}||_F^2 + \alpha||\mathbf{A}||_1 + \frac{\beta}{2} Tr(\mathbf{A}\mathbf{L}\mathbf{A}^\top), \quad ||\mathbf{u}_i||_2^2 \leq 1, \forall i,
$$

(3)

where the Laplacian term $Tr(\mathbf{A}\mathbf{L}\mathbf{A}^\top)$ is introduced to guarantee the sparse coefficients can capture the geometric structure of the data. $\mathbf{L}$ is the graph Laplacian matrix constructed from the label vector $Y = [y_1, y_2, ..., y_n] \in \mathbb{R}^n$. $\lambda$, $\alpha$ and $\beta$ are three regularization parameters.

In our bilevel model (3), the upper level feeds the representation $(\mathbf{a}_i, \mathbf{a}_j, \mathbf{a}_k)$ of the triplet constraint $(\mathbf{x}_i, \mathbf{x}_j, \mathbf{x}_k)$ into the LMNN model and directly minimizes the loss function. The lower level tries to capture the intrinsic data structure. Note that the Laplacian matrix $\mathbf{L}$ is constructed in a supervised way, thus the data structure can be well preserved even if there exists noise in the data. By solving the above optimization problem (3), a recognition-driven dictionary $\mathbf{U}$ can be learnt and accordingly leading to a well representative sparse coefficients $\mathbf{A}$. In the meantime, we can also obtain a good Mahalanobis distance matrix $\mathbf{M}$ with the new discriminative feature $\mathbf{A}$.

It is worth mentioning that the sparsity penalty and Laplacian regularization encourage the group sparsity of coefficients, thus the samples from the same class are forced to have similar sparse

representations and those from different classes are to have dissimilar sparse codes. For clarity, we show the original data (including eight classes) and its corresponding sparse coefficients learnt by our bilevel model in Fig. 1. The coefficients equipped with this useful property make the upper level easier to fulfill its mission which is to force the data samples from the same class to be close to each other and pushes those samples from different classes far away.

## 2.3 Optimization

We use the alternating direction method (ADM) to solve the optimization problem (3) after some delicate reformulations.

Let $\mathbf{A} = \mathbf{B} - \mathbf{C}$, where $\mathbf{B} \in \mathbb{R}^{k \times n}$ and $\mathbf{C} \in \mathbb{R}^{k \times n}$ are two nonnegative matrices such that $\mathbf{B}$ takes all the positive elements in $\mathbf{A}$ and the remaining elements of $\mathbf{B}$ are set to 0, while $\mathbf{C}$ does the same for the negative elements in $\mathbf{A}$ (after negation). Then the lower level optimization problem of model (3) can be transformed into the following problem:

$$\min_{\mathbf{Z}} \frac{1}{2}||\mathbf{X} - \mathbf{UPZ}||_F^2 + \alpha \mathbf{e}_{2k}^\top \mathbf{Z}\mathbf{e}_n + \frac{\beta}{2}Tr(\mathbf{PZLZ}^\top\mathbf{P}^\top), \quad s.t. \quad \mathbf{Z} \geq 0, \tag{4}$$

where $\mathbf{Z} = [\mathbf{B}; \mathbf{C}] \in \mathbb{R}^{2k \times n}$ and $\mathbf{P} = [\mathbf{I}, -\mathbf{I}] \in \mathbb{R}^{k \times 2k}$. Obviously, problem (4) is a convex problem, which can be replaced by its KKT conditions [23]. Then we obtain the following equivalent model:

$$\min_{\mathbf{M} \in \mathbb{S}_+^d, \mathbf{Z}, \mathbf{B}, \mathbf{U}} \quad (1-\lambda) \sum_{(i,j) \in \mathcal{S}} d_{\mathbf{P}^\top \mathbf{MP}}^2(\mathbf{z}_i, \mathbf{z}_j) + \lambda \sum_{(i,j,k) \in \mathcal{T}} [\xi + d_{\mathbf{P}^\top \mathbf{MP}}^2(\mathbf{z}_i, \mathbf{z}_j) - d_{\mathbf{P}^\top \mathbf{MP}}^2(\mathbf{z}_i, \mathbf{z}_k)]_+$$

$$s.t. \quad \mathbf{P}^\top \mathbf{U}^\top \mathbf{UPZ} - \mathbf{P}^\top \mathbf{U}^\top \mathbf{X} + \alpha\mathbf{E} + \beta\mathbf{P}^\top \mathbf{PZL} + \mathbf{B} = 0, \tag{5}$$

$$\mathbf{B} \odot \mathbf{Z} = 0, \quad \mathbf{Z} \geq 0, \quad \mathbf{B} \leq 0, \quad ||\mathbf{u}_i||_2^2 \leq 1, \quad \forall i \in \{1, 2, ..., k\}.$$

where $\mathbf{B} \in \mathbb{R}^{2k \times n}$ is the Lagrange multiplier matrix and $\mathbf{B}$ satisfies the constraint $\mathbf{B} \leq 0$. $\mathbf{E} \in \mathbb{R}^{2k \times n}$ is an all-one matrix.

With all these steps, the proposed bilevel distance metric learning model (3) is reformulated to a unilevel optimization problem which can be solved by ADM. We introduce two auxiliary variables $\mathbf{W}$ and $\mathbf{S}$ and relax (5) to the following problem:

$$\min_{\mathbf{M} \in \mathbb{S}_+^d, \mathbf{Z}, \mathbf{B}, \mathbf{W}, \mathbf{S}, \mathbf{U}} (1-\lambda) \sum_{(i,j) \in \mathcal{S}} d_{\mathbf{P}^\top \mathbf{MP}}^2(\mathbf{z}_i, \mathbf{z}_j) + \lambda \sum_{(i,j,k) \in \mathcal{T}} [\xi + d_{\mathbf{P}^\top \mathbf{MP}}^2(\mathbf{z}_i, \mathbf{z}_j) - d_{\mathbf{P}^\top \mathbf{MP}}^2(\mathbf{z}_i, \mathbf{z}_k)]_+$$

$$s.t. \quad \mathbf{P}^\top \mathbf{U}^\top \mathbf{UPZ} - \mathbf{P}^\top \mathbf{U}^\top \mathbf{X} + \alpha\mathbf{E} + \beta\mathbf{WL} + \mathbf{B} = 0, \mathbf{B} \odot \mathbf{S} = 0, \mathbf{P}^\top \mathbf{PZ} - \mathbf{W} = 0,$$

$$\mathbf{Z} - \mathbf{S} = 0, \quad \mathbf{S} \geq 0, \quad \mathbf{B} \leq 0, \quad ||\mathbf{u}_i||_2^2 \leq 1, \quad \forall i \in \{1, 2, ..., k\}. \tag{6}$$

The augmented Lagrangian function of problem (6) is:

$$L(\mathbf{Z}, \mathbf{B}, \mathbf{W}, \mathbf{S}, \mathbf{U}, \mathbf{M}, \mathbf{R}_1, \mathbf{R}_2, \mathbf{R}_3, \mathbf{R}_4, \mu)$$

$$=(1-\lambda) \sum_{(i,j) \in \mathcal{S}} d_{\mathbf{P}^\top \mathbf{MP}}^2(\mathbf{z}_i, \mathbf{z}_j) + \lambda \sum_{(i,j,k) \in \mathcal{T}} [\xi + d_{\mathbf{P}^\top \mathbf{MP}}^2(\mathbf{z}_i, \mathbf{z}_j) - d_{\mathbf{P}^\top \mathbf{MP}}^2(\mathbf{z}_i, \mathbf{z}_k)]_+$$

$$+\langle \mathbf{R}_1, \mathbf{P}^\top \mathbf{U}^\top \mathbf{UPZ} - \mathbf{P}^\top \mathbf{U}^\top \mathbf{X} + \alpha\mathbf{E} + \beta\mathbf{WL} + \mathbf{B}\rangle + \langle \mathbf{R}_2, \mathbf{B} \odot \mathbf{S}\rangle + \langle \mathbf{R}_3, \mathbf{P}^\top \mathbf{PZ} - \mathbf{W}\rangle \tag{7}$$

$$+\langle \mathbf{R}_4, \mathbf{Z} - \mathbf{S}\rangle + \frac{\mu}{2}||\mathbf{P}^\top \mathbf{U}^\top \mathbf{UPZ} - \mathbf{P}^\top \mathbf{U}^\top \mathbf{X} + \alpha\mathbf{E} + \beta\mathbf{WL} + \mathbf{B}||_F^2$$

$$+\frac{\mu}{2}(||\mathbf{B} \odot \mathbf{S}||_F^2 + ||\mathbf{P}^\top \mathbf{PZ} - \mathbf{W}||_F^2 + ||\mathbf{Z} - \mathbf{S}||_F^2),$$

where $\mathbf{R}_1 \sim \mathbf{R}_4$ are Lagrange multipliers, and $\mu \geq 0$ is the penalty parameter.

We alternately update the variables $\mathbf{Z}$, $\mathbf{B}$, $\mathbf{W}$, $\mathbf{S}$, $\mathbf{U}$ and $\mathbf{M}$ in each iteration by minimizing the augmented Lagrangian function of problem (6) with other variables fixed. We initialize the Mahalanobis distance matrix $\mathbf{M}$ as a unit matrix. The initialization processes of the dictionary $\mathbf{U}$ and the coefficients $\mathbf{A}$ are same as in FDDL [21]. More specifically, the iterations go as follows:

**Step 1**: Update $\mathbf{Z}$ by fixing $\mathbf{B}$, $\mathbf{W}$, $\mathbf{S}$, $\mathbf{U}$ and $\mathbf{M}$. For each $\mathbf{z}_i \in \mathbf{Z}$, we have

$$\mathbf{z}_i = \mathbf{G}_1^{-1}(\mathbf{q}_i + (1-\lambda)\mathbf{P}^\top \mathbf{MP} \sum_{(i,j) \in \mathcal{S}} \mathbf{z}_j + \lambda\mathbf{P}^\top \mathbf{MP} \sum_{(i,j,k) \in \mathcal{T}} (\mathbf{z}_j - \mathbf{z}_k)), \tag{8}$$

where $\mathbf{G}_1 = 2\mu\mathbf{P}^\top\mathbf{U}^\top\mathbf{U}\mathbf{U}^\top\mathbf{U}\mathbf{P} + 2\mu\mathbf{P}^\top\mathbf{P} + \mu\mathbf{I} + (1-\lambda)\sum_{(i,j)\in\mathcal{S}}\mathbf{P}^\top\mathbf{M}\mathbf{P}$. $\mathbf{q}_i$ is the $i$-th column of $\mathbf{Q}$, $\mathbf{Q} = \mu\mathbf{P}^\top\mathbf{U}^\top\mathbf{U}\mathbf{P}(\mathbf{P}^\top\mathbf{U}^\top\mathbf{X} - \alpha\mathbf{E} - \beta\mathbf{W}\mathbf{L} - \mathbf{B} - \mathbf{R}_1/\mu) - \mathbf{P}^\top\mathbf{P}(\mathbf{R}_3 - \mu\mathbf{W}) - \mathbf{R}_4 + \mu\mathbf{S}$.

**Step 2**: Update $\mathbf{B}$ by fixing $\mathbf{Z}$, $\mathbf{W}$, $\mathbf{S}$, $\mathbf{U}$ and $\mathbf{M}$.

$$\mathbf{B} = -\Pi_+\left((\mathbf{S}\odot\mathbf{R}_2/\mu + \mathbf{G}_2 + \beta\mathbf{W}\mathbf{L} + \mathbf{R}_1/\mu)\oslash(\mathbf{S}\odot\mathbf{S}+\mathbf{E})\right), \tag{9}$$

where $\mathbf{G}_2 = \mathbf{P}^\top\mathbf{U}^\top\mathbf{U}\mathbf{P}\mathbf{Z} - \mathbf{P}^\top\mathbf{U}^\top\mathbf{X} + \alpha\mathbf{E}$. $\Pi_+(\cdot)$ is an operator that projects a matrix onto the nonnegative cone, which can be defined as follows:

$$\Pi_+(\mathbf{X}_{ij}) = \begin{cases} \mathbf{X}_{ij}, & \text{if } \mathbf{X}_{ij} \geq 0; \\ 0, & \text{otherwise.} \end{cases} \tag{10}$$

**Step 3**: Update $\mathbf{W}$ by fixing $\mathbf{Z}$, $\mathbf{B}$, $\mathbf{S}$, $\mathbf{U}$ and $\mathbf{M}$.

$$\mathbf{W} = \left[\mathbf{P}^\top\mathbf{P}\mathbf{Z} + \mathbf{R}_3/\mu - \beta(\mathbf{G}_2 + \mathbf{B} + \mathbf{R}_1/\mu)\mathbf{L}^\top\right]\left(\beta^2\mathbf{L}\mathbf{L}^\top + \mathbf{I}\right)^{-1}. \tag{11}$$

**Step 4**: Update $\mathbf{S}$ by fixing $\mathbf{Z}$, $\mathbf{B}$, $\mathbf{W}$, $\mathbf{U}$ and $\mathbf{M}$.

$$\mathbf{S} = \Pi_+\left((\mathbf{Z} + \mathbf{R}_4/\mu - \mathbf{B}\odot\mathbf{R}_2/\mu)\oslash(\mathbf{B}\odot\mathbf{B}+\mathbf{E})\right). \tag{12}$$

**Step 5**: Update $\mathbf{U}$ by fixing $\mathbf{Z}$, $\mathbf{B}$, $\mathbf{W}$, $\mathbf{S}$ and $\mathbf{M}$. We need to solve the following problem:

$$\mathbf{U} = \arg\min_{\mathbf{U}\in\Omega} \quad ||\mathbf{G}_2 + \beta\mathbf{W}\mathbf{L} + \mathbf{B} + \mathbf{R}_1/\mu||_F^2, \tag{13}$$

where $\Omega = \{\mathbf{U} \mid ||\mathbf{U}_i||_2^2 \leq 1, i = 1, ..., k\}$. The problem (13) is a quartic polynomial minimization problem. It is difficult to compute its exact solution. So we use the projected gradient descent method to update $\mathbf{U}$:

$$\mathbf{U} = \Pi_\Omega(\mathbf{U} - \eta_1\nabla_\mathbf{U}), \tag{14}$$

with $\nabla_\mathbf{U} = 2\left(\mathbf{U}(\mathbf{G}_4^\top + \mathbf{G}_4) + \mathbf{U}(\mathbf{G}_5^\top + \mathbf{G}_5)\right) - 4\left(\mathbf{U}(\mathbf{G}_6^\top + \mathbf{G}_6) + \mathbf{G}_7^\top\right) + 2\mathbf{U}(\mathbf{G}_8^\top + \mathbf{G}_8) - 2\mathbf{X}\mathbf{G}_3^\top\mathbf{P}^\top + 4\mathbf{X}\mathbf{X}^\top\mathbf{U}$, where $\mathbf{G}_3 = \alpha\mathbf{E} + \beta\mathbf{W}\mathbf{L} + \mathbf{B} + \mathbf{R}_1/\mu$, $\mathbf{G}_4 = \mathbf{P}\mathbf{Z}\mathbf{Z}^\top\mathbf{P}^\top\mathbf{U}^\top\mathbf{U}$, $\mathbf{G}_5 = \mathbf{U}^\top\mathbf{U}\mathbf{P}\mathbf{Z}\mathbf{Z}^\top\mathbf{P}^\top$, $\mathbf{G}_6 = \mathbf{P}\mathbf{Z}\mathbf{X}^\top\mathbf{U}$, $\mathbf{G}_7 = \mathbf{U}^\top\mathbf{U}\mathbf{P}\mathbf{Z}\mathbf{X}^\top$, $\mathbf{G}_8 = \mathbf{P}\mathbf{Z}\mathbf{G}_1^\top\mathbf{P}^\top$. $\Pi_\Omega(\mathbf{U})$ is the projection of the matrix $\mathbf{U}$ onto $\Omega$ and $\eta_1$ is a step size.

**Step 6**: Update $\mathbf{M}$ by fixing $\mathbf{Z}$, $\mathbf{B}$, $\mathbf{W}$, $\mathbf{S}$ and $\mathbf{U}$. The objective function is linear with respect to $\mathbf{M}$, we directly adopt subgradient descent to update $\mathbf{M}$ in each iteration. As before, set $\mathbf{z}_{ij} = \mathbf{z}_i - \mathbf{z}_j$, then the subgradient of problem (6) with respect to $\mathbf{M}$ can be calculated as follows:

$$\nabla_\mathbf{M} = (1-\lambda)\sum_{(i,j)\in\mathcal{S}}\mathbf{P}\mathbf{z}_{ij}\mathbf{z}_{ij}^\top\mathbf{P}^\top + \lambda\sum_{(i,j,k)\in\mathcal{T}^+}(\mathbf{P}\mathbf{z}_{ij}\mathbf{z}_{ij}^\top\mathbf{P}^\top - \mathbf{P}\mathbf{z}_{ik}\mathbf{z}_{ik}^\top\mathbf{P}^\top), \tag{15}$$

where $\mathcal{T}^+$ denotes the subset of constraints in $\mathcal{T}$ that is larger than 0 in function (6). After each iteration, $\mathbf{M}$ is projected onto the positive semidefinite cone:

$$\mathbf{M} = \Pi_{\mathbb{S}_+^d}(\mathbf{M} - \eta_2\nabla_\mathbf{M}), \tag{16}$$

where $\eta_2$ is a step size, and $\Pi_{\mathbb{S}_+^d}(\mathbf{M})$ is the orthogonal projection of the matrix $\mathbf{M}\in\mathbb{S}^d$ onto the positive semidefinite cone $\mathbb{S}_+^d$. The specific procedures are summarized in Algorithm 1.

## 2.4 Classification Scheme

When problem (6) is solved, we obtain a dictionary $\mathbf{U}$ and the sparse coefficients $\mathbf{A} = \mathbf{P}\mathbf{Z}$ of training samples. In the testing phase, given a testing sample $\mathbf{x}$, we first compute its sparse coefficient by the vector form of the lower level optimization model:

$$\mathbf{a}^* = \arg\min_\mathbf{a}\frac{1}{2}||\mathbf{x} - \mathbf{U}\mathbf{a}||_F^2 + \alpha||\mathbf{a}||_1 + \frac{\beta}{2}\sum_{i\in N_s(\mathbf{x})}q_i||\mathbf{a} - \mathbf{a}_i||_2^2, \tag{17}$$

where $N_s(\mathbf{x})$ denotes the set of $s$ nearest neighbors of $\mathbf{x}$ and the $s$ nearest neighbors are chosen from training samples $\mathbf{X}$. $\mathbf{a}_i$ is the coefficient of the $i$-th training sample $\mathbf{x}_i$. $q_i$ is the weight between the

---

**Algorithm 1** Algorithm to solve Eq. (6)

---

1: **Input:** $\mathcal{S}, \mathcal{T}, \mathbf{X} \in \mathbb{R}^{d \times n}, \mathbf{L}, \lambda, \alpha, \beta$
2: **Output:** $\mathbf{M} \in \mathbb{S}_+^d, \mathbf{U}, \mathbf{A}$
3: **Initialization:** $\mathbf{M}^0, \mathbf{U}^0, \mathbf{A}^0, \mathbf{Z}^0 = \mathbf{P}^\dagger \mathbf{A}^0, \mathbf{S}^0 = \mathbf{Z}^0, \mathbf{W}^0 = \mathbf{P}^\top \mathbf{P} \mathbf{Z}^0$, and $\mathbf{B}^0 = \mathbf{P}^\top (\mathbf{U}^0)^\top \mathbf{X} - \mathbf{P}^\top (\mathbf{U}^0)^\top \mathbf{U}^0 \mathbf{P} \mathbf{Z}^0 - \alpha \mathbf{E} - \beta \mathbf{W}^0 \mathbf{L}$. Set $\mathbf{R}_1 = \mathbf{0}_d, \mathbf{R}_2 = \mathbf{0}_d, \mathbf{R}_3 = \mathbf{0}_d$, $\mathbf{R}_4 = \mathbf{0}_d, \mu_0 = 1e - 3, \mu_{max} = 1e + 8, \rho = 1.3, \varepsilon_1 = 1e - 4, \varepsilon_2 = 1e - 5$, and $t = 0$.
4: **repeat**
5:     Steps 1∼6;
6:     Update Lagrange multipliers and $\mu^{t+1}$:
        $\mathbf{R}_1^{t+1} = \mathbf{R}_1^t + \mu \left( \mathbf{P}^\top (\mathbf{U}^{t+1})^\top \mathbf{U}^{t+1} \mathbf{P} \mathbf{Z}^{t+1} - \mathbf{P}^\top (\mathbf{U}^{t+1})^\top \mathbf{X} + \alpha \mathbf{E} + \beta \mathbf{W}^{t+1} \mathbf{L} + \mathbf{B}^{t+1} \right)$,
        $\mathbf{R}_2^{t+1} = \mathbf{R}_2^t + \mu(\mathbf{B}^{t+1} \odot \mathbf{S}^{t+1}), \mathbf{R}_3^{t+1} = \mathbf{R}_3^t + \mu(\mathbf{P}^\top \mathbf{P} \mathbf{Z}^{t+1} - \mathbf{W}^{t+1})$,
        $\mathbf{R}_4^{t+1} = \mathbf{R}_4^t + \mu(\mathbf{Z}^{t+1} - \mathbf{S}^{t+1}), \mu^{t+1} = \min(\rho \mu^t, \mu_{max})$;
7:     $t \leftarrow t + 1$;
8: **until** Converge

---

training sample $\mathbf{x}_i$ and the test sample $\mathbf{x}$. Note that in the training phase, we construct the weight matrix $\mathbf{Q}$ as follows:

$$Q_{ij} = \begin{cases} 1, & \text{if samples } \mathbf{x}_i \text{ and } \mathbf{x}_j \text{ belong to the same class}, \\ 0, & \text{otherwise}. \end{cases} \tag{18}$$

Then we compute the corresponding Laplacian matrix $\mathbf{L} = \mathbf{T} - \mathbf{Q}$, where $\mathbf{T}$ is a diagonal matrix and $T_{ii} = \sum_j Q_{ij}$. In the testing phase, we find $s$ nearest neighbors from training set for each test sample. In the experiment, we set $s = 5$ and the weight $q_i = 1$ ($\forall i \in N_s(\cdot)$).

After the coefficient $\mathbf{a}^*$ of the test sample $\mathbf{x}$ is obtained, the squared Mahalanobis distance between the test sample $\mathbf{x}$ and the training sample $\mathbf{x}_i$ can be calculated as:

$$d_\mathbf{M}^2(\mathbf{a}^*, \mathbf{a}_i) = (\mathbf{a}^* - \mathbf{a}_i)^\top \mathbf{M} (\mathbf{a}^* - \mathbf{a}_i), \tag{19}$$

where $\mathbf{M}$ is the learned optimal distance matrix. The test sample $\mathbf{x}$ is then classified to the class where its nearest training sample belongs.

## 2.5 Convergence Analysis

There are lots of researchers focusing on the convergence of ADM with two blocks of variables. However, there is still no affirmative convergence proof for multi-block convex minimization problem where the objective function consists of more than two separable convex functions. The recent solution is to use an additional dual step-size parameter $\mu$ in updating Lagrange multipliers (as shown in Algorithm 1). This scheme is simple and effective because it not only requires no additional assumptions associated with the objective function but also guarantees the convergence of ADM with multi-block variable under mild assumptions [9]. For these reasons, it is enough that we only need to choose a proper step-size parameter $\mu$ and termination conditions.

To further illustrate the convergence of ADM in solving the proposed model (6), we conduct several experiments on three datasets, including NUST Robust Face database (NUST-RF) [2], OSR dataset [13] and PubFig database [6]. Note that there are two environments in NUST-RG database, *i.e.*, indoor and outdoor. The objective function values versus number of iterations are shown in Fig. 2. From the figure we can see that the objective values reduce reasonably well.

## 3 Experimental Results

We evaluate the proposed algorithm over different classification databases, including real-world malicious occlusion datasets, contiguous occlusion and corruption datasets. There are two main goals in our experiments: first, we will show that our bilevel model is more robust to be applied to solve real-world occlusion problems; second, our model is able to outperform the related metric learning methods.

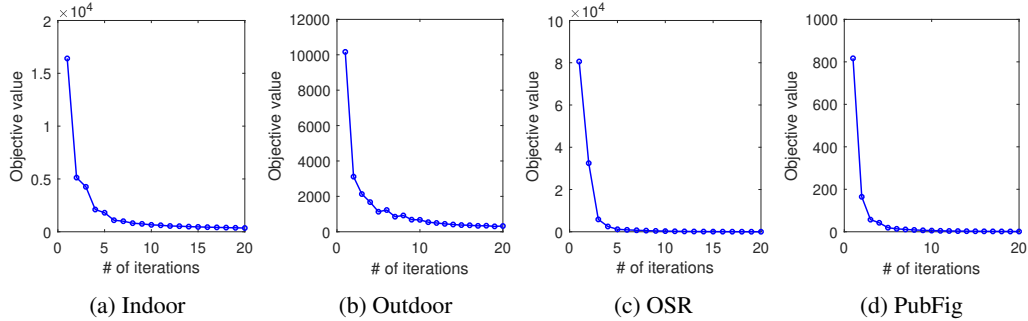

| (a) Indoor | (b) Outdoor | (c) OSR | (d) PubFig |

Figure 2: Objective value vs. the number of iterations.

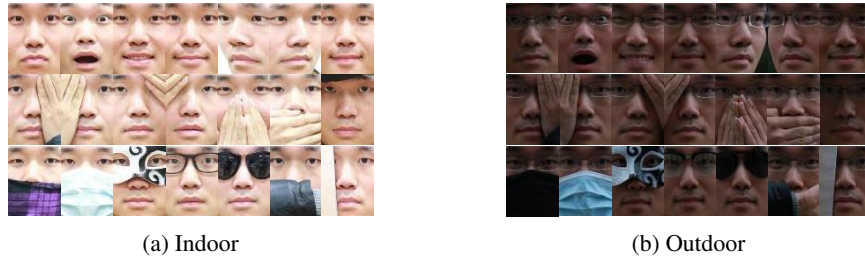

| (a) Indoor | (b) Outdoor |

Figure 3: Cropped images of one subject captured in two environments in NUST-RF database, *i.e.*, (a) indoor, and (b) outdoor.

## 3.1 Compared Methods

We compare the proposed bilevel distance metric learning model with the following methods: the base-line KNN [14], LMNN [16], FANTOPE [7], CAP [5] and RML [10]. Specifically, as baselines, we consider the most relevant technique from the literature, *i.e.*, $k$-nearest neighbor method (KNN). KNN computes Euclidean distance to measure the similarity between any two images. LMNN is one of the most widely-used Mahalanobis distance metric learning methods, which uses labeled information to generate triplet constraints. FANTOPE method is based on LMNN, and it utilizes a fantope regularization which minimizes sum of $k$ smallest singular values of distance matrix $\mathbf{M}$. Same as FANTOPE method, CAP method is also based on LMNN, and it uses a capped trace norm to penalize the singular values of distance matrix $\mathbf{M}$ that are less than a threshold adaptively learned in the optimization. RML learns the discriminative distance matrix by enforcing a margin between the inter-class sparse reconstruction residual and intra-class sparse reconstruction residual.

For all metric learners, we use 5-fold cross validation and gauge the average accuracy and standard deviation as final performance. All the regularization parameters are tuned from range $\{10^{-4}, 10^{-3}, 10^{-2}, 10^{-1}, 1, 10, 10^2\}$. For CAP and FANTOPE methods, the parameter rank of distance matrix $\mathbf{M}$ is tuned from $[10 : 5 : 30]$. For a fair comparison, we specify 1 "target" neighbor for each training sample for all LMNN related methods. In testing phase, we use 1-NN method.

## 3.2 Real-World Malicious Occlusion

First we consider the NUST Robust Face database (NUST-RF) [2]. It is mainly designed for robust face recognition under various occlusions. Except occlusion, it also includes variations of illumination, expression and pose. We use a subset face images of NUST-RF database, and there are 50 subjects captured in two environments (indoor and outdoor). We manually crop the face portion of the image and then normalize it to $80 \times 60$ pixels. Fig. 3 shows an example of several selected images of one subject. We extracted LOMO features for each image [8], which not only achieve some invariance to viewpoint changes, but also capture local region characteristics of a person. PCA is further applied to reduce the feature dimension to 30.

Table 1 shows the recognition performance of different methods on NUST-RF database of two environments. Obviously, our method outperforms other competing methods in indoor case and gets

Table 1: Recognition accuracy (%) and standard deviation of different methods on NUST-RF database in two environments.

|  | KNN [14] | LMNN [16] | FANTOPE [7] | CAP [5] | RML [10] | Proposed |
|---|---|---|---|---|---|---|
| Indoor | $36.14 \pm 2.70$ | $36.20 \pm 3.30$ | $41.87 \pm 2.50$ | $41.70 \pm 2.86$ | $35.56 \pm 3.04$ | $\mathbf{47.84 \pm 1.18}$ |
| Outdoor | $45.24 \pm 1.51$ | $46.01 \pm 2.06$ | $\mathbf{58.72 \pm 1.33}$ | $58.34 \pm 1.33$ | $42.81 \pm 2.16$ | $58.21 \pm 1.68$ |

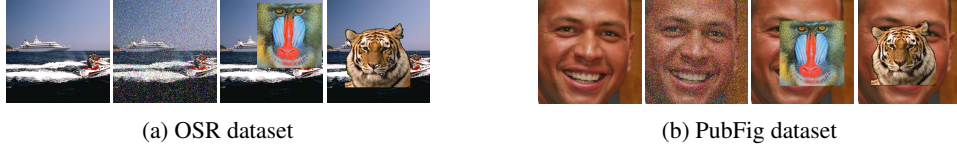

(a) OSR dataset             (b) PubFig dataset

Figure 4: Example pairs of images from two datasets, *i.e.*, (a) OSR dataset, (b) PubFig dataset. For each subfigure, from left to right: original image, its noisy versions (with sparse noise, regular occlusion and irregular occlusion, respectively).

comparable results with FANTOPE method in outdoor case. This is because the proposed model jointly extracts features under a sparse-representation model and performs distance metric learning task at the same time. In this way, our features are more robust to noise, thus we can get better results than LMNN which is only based on extracted features. For FANTOPE and CAP methods, they also achieve relatively good results because the low-rank regularization on $\mathbf{M}$ fits this face database just right. Moreover, as shown in Fig. 3, if occlusions exist, it is unlikely that the test image will be very close to any single training image of the same class, so that the KNN classifier performs poorly. Although LMNN can improve the recognition rates compared to KNN, their improvements are limited. RML also performs poorly because it is based on the MSE criterion which is sensitive to outliers.

### 3.3 Sparse Noise and Contiguous Occlusion

Next we did three groups of occlusion experiments associated with two datasets, *i.e.,* OSR dataset [13] and PubFig database [6], to validate the robustness of the proposed algorithm. There are 2688 images from 8 scene categories in OSR dataset. We extract gist features as representation [12]. For PubFig database, we use a subset face images and there are 771 images from 8 face categories [6]. Similarly with NUST-RF database, we extract LOMO features as representation. We simulate various types of contiguous occlusion by adding sparse noise to both training and testing data or by replacing a randomly selected local region in each image with an unrelated square block of the "baboon" image for regular occlusion and a randomly located "tiger" image for irregular occlusion. Sparse noise is simulated by 20 adding Gaussian noise with zero mean and 0.01 variance to both training and testing data. And the size of the added image is 60% of the size of the original image. Fig. 4 shows a clean image and its noisy versions from two datasets. Since the differences between the pixels of the unrelated "baboon" image or "tiger" image and the pixels of the images from two datasets are relatively small, the contiguous occlusion caused by these unrelated images is much more challenging than by random black or white dots.

Table 2 and Table 3 show the classification accuracy and the standard derivation of different methods on two datasets, *i.e.*, OSR dataset and PubFig dataset. It is obvious our method consistently outperforms other competing methods in most cases, especially on the occlusion data. This is because our bilevel model jointly performs metric learning and extracts features at the same time. And since we use the sparsity penalty and graph regularization in the lower level model, the new features is not only more robust to noise but also discriminative. For this classification task, both FANTOPE and CAP methods are based on LMNN method. Since they all have similar results, which indicates the low-rank regularization on $\mathbf{M}$ for Mahalanobis distance metric learning is not particularly effective in this case. Especially for regular occlusion that replaces a randomly selected local region with "baboon" image and irregular occlusion that replaces local region with "tiger" image, LMNN, FANTOPE and CAP achieve almost the same result. For RML method, due to the limitation that RML is based on the MSE criterion, it still performs poorly.

Table 2: Recognition accuracy (%) and standard deviation of different methods on OSR dataset, where sparse noise, regular and irregular occlusions are added.

|  | KNN [14] | LMNN [16] | FANTOPE [7] | CAP [5] | RML [10] | Proposed |
|---|---|---|---|---|---|---|
| Original | $69.01 \pm 1.96$ | $74.41 \pm 1.20$ | $\mathbf{74.97 \pm 0.88}$ | $74.45 \pm 1.19$ | $61.34 \pm 1.62$ | $74.43 \pm 1.14$ |
| Sparse | $61.83 \pm 1.75$ | $66.67 \pm 1.70$ | $66.70 \pm 1.68$ | $66.67 \pm 1.70$ | $56.57 \pm 2.60$ | $\mathbf{68.72 \pm 2.72}$ |
| Regular | $55.34 \pm 2.72$ | $58.66 \pm 1.31$ | $58.73 \pm 1.43$ | $58.70 \pm 1.27$ | $54.38 \pm 3.26$ | $\mathbf{64.77 \pm 1.46}$ |
| Irregular | $52.25 \pm 1.74$ | $57.02 \pm 1.80$ | $57.10 \pm 1.74$ | $57.13 \pm 1.68$ | $50.45 \pm 2.17$ | $\mathbf{62.72 \pm 3.06}$ |

Table 3: Recognition accuracy (%) and standard deviation of different methods on PubFig dataset, where sparse noise, regular and irregular occlusions are added.

|  | KNN [14] | LMNN [16] | FANTOPE [7] | CAP [5] | RML [10] | Proposed |
|---|---|---|---|---|---|---|
| Original | $56.73 \pm 1.12$ | $61.65 \pm 1.63$ | $61.69 \pm 1.60$ | $61.80 \pm 1.72$ | $55.86 \pm 1.54$ | $\mathbf{63.46 \pm 1.65}$ |
| Sparse | $48.46 \pm 1.35$ | $51.35 \pm 1.55$ | $51.39 \pm 1.69$ | $51.39 \pm 1.99$ | $48.23 \pm 1.26$ | $\mathbf{54.40 \pm 1.59}$ |
| Regular | $35.30 \pm 1.14$ | $37.48 \pm 1.64$ | $37.71 \pm 1.56$ | $38.05 \pm 1.57$ | $35.80 \pm 1.59$ | $\mathbf{44.29 \pm 0.54}$ |
| Irregular | $40.94 \pm 2.30$ | $41.73 \pm 3.39$ | $41.88 \pm 2.78$ | $42.33 \pm 2.35$ | $40.23 \pm 2.48$ | $\mathbf{49.10 \pm 1.09}$ |

To discuss the influences of individual parameters on the performance of the proposed model, we take PubFig dataset with regular occlusion as an example. We test the influence of parameters $\lambda$, $\alpha$, $\beta$ on the recognition accuracy as shown in Fig. 5.

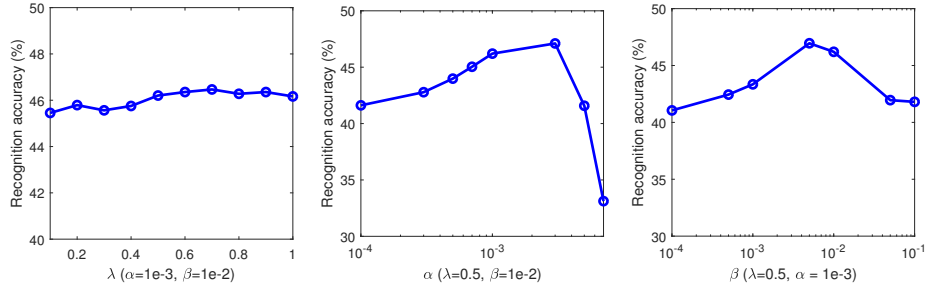

Figure 5: The influence of parameters $\lambda$, $\alpha$, $\beta$ on the recognition accuracy of PubFig dataset with regular occlusion.

## 4 Conclusion

We propose a new bilevel distance metric learning model for robust image recognition task. Different from conventional metric learning methods which learn a Mahalanobis distance matrix based on extracted features, we dig the intrinsic data structures using the Laplacian graph regularized sparse coefficients and jointly perform distance metric learning at the same time. Due to the feature extraction operation of the lower level model, the new descriptors become more robust to noise with the sparsity norm and more discriminative with the Laplacian graph term, leading to good recognition performance. Moreover, we also derive an efficient algorithm to solve the proposed new model. Extensive experiments on several occluded datasets verify the remarkable performance improvements led by the proposed bilevel model.

### Acknowledgments

J.X. and C.D. were partially supported by the National Natural Science Foundation of China 61572388, the National Key Research and Development Program of China (2017YFE0104100), and the Key R&D Program-The Key Industry Innovation Chain of Shaanxi under Grants 2017ZDCXL-GY-05-04-02 and 2018ZDXM-GY-176.
L.L. and H.H. were partially supported by U.S. NSF-IIS 1836945, NSF-IIS 1836938, NSF-DBI 1836866, NSF-IIS 1845666, NSF-IIS 1852606, NSF-IIS 1838627, NSF-IIS 1837956.

## Footnotes

*J.X. and L.L. made equal contributions. C.D. and H.H. are corresponding authors.

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
