[Reviews · NeurIPS 2018]

Reviewer 1



In this manuscript, authors propose to learn the low level features and high level features simultaneously. By learning the sparse low level features, authors claim that they are more robust for the metric learning process. My concerns are as follows. 1. The proposed method may be inefficient for the real-world applications. Both number of examples and dimensionality of features in the experiments are small. Note that when updating $Z$, authors has to enumerate triplet constraints, whose size is cubic in the number of training data. When projecting the updated metric back to the PSD cone, the cost is cubic in the number of low level features. It makes the algorithm hard to handle the real applications. For the test phase, the proposed algorithm also has to obtain the sparse code for each example. 2. Authors didn't mention the size of dictionary in experiments. The size of metric is quadratic in the number of low level features. If the size of dictionary is too large, which is ubiquitous for image classification, the proposed method can be impractical. Besides, authors should report more settings in experiments, e.g., $k$ in k-NN, number of iterations of LMNN, $k$ in LMNN, etc. 3. For the problem in Eqn.6, it can be nonconvex since both of metric M and Z are variables to be solved. So the analysis in Section 2.5 is suspect. After the rebuttal: 1. Authors applies 1-NN to alleviate the large-scale problem, which is not convincing. 2. The setting in the experiments is not common for DML. Besides, codebook with size of 120 is too small for a meaningful sparse coding.

Reviewer 2



This paper is not in my area of expertise.

Reviewer 3



This paper proposes a bilevel distance metric learning method for robust image recognition task. A set of experiments validate that the proposed method is competitive with the state-of-art algorithms on the robust image recognition task. The main idea of this paper is novel. Based on my knowledge, this is the first work on bilevel metric learning. In the proposed bilevel model, the lower level characterizes the intrinsic data structure using graph regularized sparse coefficients, while the upper level forces the data samples from the same class to be close to each other and simultaneously pushes those from different classes far away. Thus, the proposed model combines the advantages of metric learning and dictionary learning. The proposed bilevel model is optimized via ADMM method. During the optimization step, the authors converted the lower level constraint in Eq. (3) into an equivalent one in Eq. (5) using KKT conditions. This is a good and interesting idea. This paper is well-organized. The motivations of the proposed model and algorithm are articulated clearly. Meanwhile, the derivations and analysis of the proposed algorithm are correct. The experiments on image classification with occlusion/noise are interesting, because few existing papers on metric learning can effectively address these difficult cases. I think the main reason is that the proposed method unites the advantages of dictionary learning models. The experimental results show that the proposed model is better than the other existing methods. A few additional comments on this paper: 1.The experimental settings should be further clarified. 2.There are some parameters in the proposed model, so the authors need to discuss the influences of individual parameters on the model’s performance. I have read the authors’ rebuttal and other reviews. I think this work is solid, so will keep my score and vote for acceptance.

Reviewer 4



Summary: The authors propose a bilevel method for metric learning, where the lower level is responsible for the extraction of discriminative features from the data based on a sparse coding scheme with graph regularization. This effectively detects their underlying geometric structure, and the upper level is a classic metric learning approach that utilizes the learned sparse coefficients. These two components are integrated into a joint optimization problem and an efficient optimization algorithm is developed accordingly. Hence, new data can be classified based on the learned dictionary and the corresponding metric. In the experiments the authors demonstrate the capabilities of the model to provide more discriminative features from high dimensional data, while being more robust to noise. Quality: The technical contribution of the paper seems to be pretty solid. In particular, the argumentation about the feature extraction and why this is beneficial for the metric learning, is pretty convincing. Also, the final objective function is directly interpretable. In addition, the derivation of the optimization algorithm is as clean as possible, and nicely presented. Furthermore, the converge analysis of the algorithm is discussed as well. So the proposed framework is sufficiently complete. However, I have two comments: 1) The graph regularizer in Eq.16 utilizes the weights q_i, which are not defined properly in the text. Especially, given that the Laplacian is constructed during training in a supervised way. 2) Could you provide some information considering the efficiency of the model? Consider for instance the training time. Clarity: The paper is very well-written and nicely organized. Also, it provides neatly all the necessary information to the reader. The related work seems to be discussed properly. I have few minor comments: 1) In Fig.1 it would be nice if you could give a better explanation of the x-axis. To my undestanding these are the samples i=1,...n categorized, but this is not direcly communicated by the figure. 2) Does the construction of the Laplacian in a supervised way implies that you construct the adjacency matrix based on the given labels? 3) In Fig.2 you could have used log-scale for the y-axis. Originality: I am not an expert in the field, but the related work seems to be properly cited. Also, by taking into account the related work as presented by the authors, the proposed model seems to be novel enough. Even if the bilevel concept with the graph regularized sparse coding as an integrated feature extractor has been utilized in previous models, in the current paper it is considered into the metric learning framework. Also, the provided optimization algorithm has been developed specifically for the proposed model. Significance: The idea of integrating a meaningful feature extractor in the metric learning task, is very interesting and makes a lot of sense. Especially, I like the fact that in the proposed model this combination is implemented naturally. Also, the developed optimization algorithm is remarkable, since it solves a rather complicated problem. The improvement upon the results of the experiments is considerable, which shows that the proposed approach is particularly useful. General comments: The general idea of bilevel models is pretty appealing. In particular, the lower level is designed in such a way that it provides meaningful features into the upper level, which solves the main problem. In this paper, the discriminative features that capture the underlying geometric structure of the data are utilized such that to learn a Mahalanobis distance for classification. The provided optimization algorithm that solves the joint problem is very interesting as well, since the problem is challenging. The experimental results show that, indeed, extracting meaningful features from the data, can lead to significant improvements. ------------------------------------------------ After reading the rebuttal my final comments are that the submission seems to be solid, and also, considering the replies of the authors, I feel that they answer sufficiently well all the comments of the reviewers. So I will keep my score and vote for acceptance.